# Comparing Representations in Static and Dynamic Vision Models to the Human Brain

**Hamed Karimi**
Department of Psychology and Neuroscience
Boston College
Massachusetts, MA 02467
`karimike@bc.edu`

**Stefano Anzellotti**
Department of Psychology and Neuroscience
Boston College
Massachusetts, MA 02467
`stefano.anzellotti@bc.edu`

## Abstract

We compared neural responses to naturalistic videos and representations in deep network models trained with static and dynamic information. Models trained with dynamic information showed greater correspondence with neural representations in all brain regions, including those previously associated with the processing of static information. Among the models trained with dynamic information, those based on optic flow accounted for unique variance in neural responses that were not captured by Masked Autoencoders. This effect was strongest in ventral and dorsal brain regions, indicating that despite the Masked Autoencoders' effectiveness at a variety of tasks, their representations diverge from representations in the human brain in the early stages of visual processing.

## 1 Introduction

The human visual system is organized into distinct processing streams [Ungerleider et al., 1982, Pitcher and Ungerleider, 2021]: a ventral stream that extends from early visual regions into the inferior portions of temporal cortex, and a dorsal stream that extends into lateral occipital cortex and branches into a lateral stream (along the superior temporal sulcus) and a parietal stream (reaching the inferior parietal lobule). This organization likely results from the computational requirements of visual perception. Therefore, understanding the representations encoded by different visual streams could offer insights about the human brain and also about more general principles of vision.

The ventral stream has been proposed to encode static object identity [Grill-Spector and Weiner, 2014], while dynamic information has been associated with the dorsal, lateral and parietal streams [Ganel and Goodale, 2003, Culham et al., 2003]. Indeed, static images of objects are known to drive responses in ventral temporal regions in macaques [Pasupathy and Connor, 2002, Logothetis et al., 1995, Tanaka, 1996, Hung et al., 2005] and in humans [Edelman et al., 1998, Haxby et al., 2001]. Moving stimuli drive stronger responses in dorsal and lateral regions [Zeki et al., 1991, Tootell et al., 1995, Saito et al., 1986]. In addition, disruption to lateral regions using TMS affects the processing of dynamic information [Beckers and Hömberg, 1992, Pitcher et al., 2014] as well as motion prediction [Vetter et al., 2015].

However, other studies have challenged the hypothesis that visual streams in the human brain differ based on whether they encode static or dynamic visual features. These studies suggested that both static and dynamic features are represented in multiple visual streams [Kourtzi et al., 2002, Freud et al., 2017, Cornette et al., 1998, Sunaert et al., 1999, Robert et al., 2023]. Here, we investigated the contribution of static and dynamic information to the representations encoded by different visual streams by quantifying the convergence between neural representations and representations learned by deep network models.

Preprint.

Previous work compared neural responses to deep network models trained with static images [Yamins et al., 2013, Khaligh-Razavi and Kriegeskorte, 2014, Zhuang et al., 2021, Konkle and Alvarez, 2022]. The present work studies the additional contribution of dynamic information during the observation of quasi-naturalistic videos by comparing neural responses to deep networks whose inputs are static images (e.g., convolutional ResNets [He et al., 2016], image masked autoencoders [He et al., 2022]) and to deep networks whose inputs are videos (e.g., hidden two-stream networks, video masked autoencoders [Zhu et al., 2019, Tong et al., 2022, Feichtenhofer et al., 2022]). Critically, we include in our analyses a family of self-supervised models that are widely used in Computer Vision but that are understudied in Cognitive Neuroscience: masked autoencoders (MAEs) [He et al., 2022].

Previous work has indicated that models yielding more accurate categorization performance also typically offer a more accurate prediction of neural responses [Yamins et al., 2014]. MAEs are remarkably effective at categorization, making them promising candidate models of neural responses. Additionally, thanks to their transformer architecture, MAEs can capture larger-scale spatial dependencies in images, and masked video autoencoders can capture temporal dependencies. Human observers are able to represent such dependencies as well, thus in principle, MAEs could account for unique variance in neural representations compared to convolutional neural networks (CNNs).

## 2 Methods

### 2.1 Data

BOLD fMRI responses (3×3×3 mm) to eight movie segments of 'Forrest Gump' were obtained from the publicly available *studyforrest* audiovisual dataset (`http://studyforrest.org`). Fifteen right-handed participants took part in the study (6 females; age range 21-39 years, mean 29.4 years). The data was acquired with a T2*-weighted echo-planar imaging sequence, using a whole-body 3 Tesla Philips Achieva dStream MRI scanner equipped with a 32 channel head coil.

### 2.2 Preprocessing

Data were first preprocessed using fMRIPrep (`https://fmriprep.readthedocs.io/en/latest/index.html`): a robust pipeline for the preprocessing of diverse fMRI data. Anatomical images were skull-stripped with ANTs (`http://stnava.github.io/ANTs/`), and FSL FAST was used for tissue segmentation. Functional images were corrected for head movement with FSL MCFLIRT (`https://fsl.fmrib.ox.ac.uk/fsl/fslwiki/MCFLIRT`), and were subsequently coregistered to their anatomical scan with FSL FLIRT. Finally, the skull-stripped anatomical images were normalized to the MNI template using SPM. We denoised the data with CompCor Behzadi et al. [2007] using 5 principal components extracted from the union of cerebrospinal fluid and white matter.

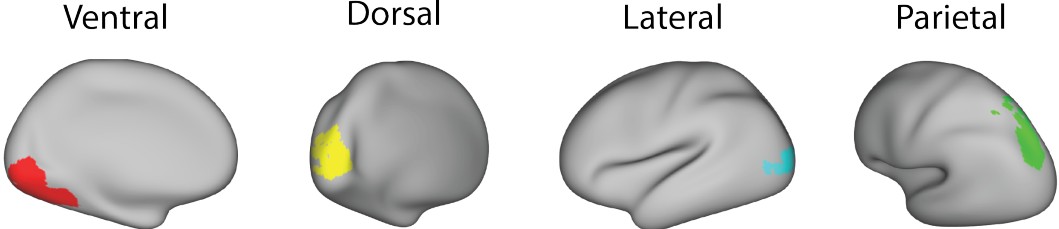

Figure 1: Masks of the visual streams in the human brain projected on an inflated cortical surface in MNI space.

### 2.3 Regions of Interest (ROI)

To identify the regions of interest (ROI), we used an atlas of probabilistic maps of visual topography in the human cortex from a previous study [Wang et al., 2015]. The atlas contains twenty-five cortical regions and spans multiple visual streams: ventral, dorsal, parietal, and lateral (Figure 1).

A list of probabilities is associated with each voxel to reflect the likelihood of that voxel being part of each of the twenty-five brain regions ($R_i, i = 1, \ldots, 25$). We calculated the transformation from MNI

Table 1: Models of visual cortex

| Model | Input | Output | Training dataset | #Selected layers |
|---|---|---|---|---|
| Supervised static | image | object identity | Image-net | 11 |
| Supervised static | image | action identity | HAA-500 | 11 |
| Self-supervised dynamic | video | optic flows | HAA-500 | 11 |
| Supervised dynamic | optic flow | action identity | HAA-500 | 11 |
| pre-trained Masked Autoencoder | (masked) image | (unmasked) image | Image-net | 12 |
| fine-tuned Masked Autoencoder | image | object identity | Image-net | 12 |
| pre-trained Masked Video Autoencoder | (masked) video | (unmasked) video | Kinetics-400 | 12 |
| fine-tuned Masked Video Autoencoder | video | action identity | Kinetics-400 | 12 |
| pre-trained Masked Video Distillation | (masked) video | MAE & VideoMAE high-level features | Kinetics-400 | 12 |

space to each participant's native space and co-registered the probability maps with each participant's anatomy. To prevent overlap between the regions of interest in the participants' native space, we followed a procedure analogous to [Wang et al., 2015]. Specifically, we calculated the maximum probability map for each participant, using which we exclusively classified each voxel as either belonging to a specific ROI or as being outside of all the ROIs.

The inclusion probability was computed as the probability of a voxel of being in any of the defined regions ($P(\cup_{i=1}^{25} v \in R_i)$), and The exclusion probability is the probability of a voxel not belonging to any of the ROIs ($P(\cap_{i=1}^{25} v \notin R_i)$). If the exclusion exceeded the inclusion probability, we discarded the voxel. Otherwise, we classified the voxel as belonging to the region with the highest probability. The resulting ROIs were grouped into four sets associated with distinct visual streams. The ventral stream contains V1v, V2v, V3v, hV4, VO1, VO2, PHC1, PHC2; the dorsal stream V1d, V2d, V3d, V3a, V3b; the lateral stream LO1, LO2, hMT, and finally the parietal stream IPS0, IPS1, IPS2, IPS3, IPS4, IPS5, SPL1, and FEF. While often the term "dorsal stream" is used to refer to the combination of the regions we labeled as "dorsal stream" and the regions we labeled as "parietal stream", here we sought to distinguish between the initial branch of the dorsal stream and its parietal and lateral temporal continuations, without implying that the initial segment is disproportionately associated with one or the other.

## 2.4 Models of human visual cortex

To study representations of quasi-naturalistic visual stimuli, we used a variety of vision models, including feed-forward convolutional neural networks, as well as state-of-the-art foundation vision models. The models vary in architecture, learning objective, and training data (Table 1). Here, we propose an overview of the models. Training details for the HAA-trained CNNs are presented in supplementary materials. The trained versions of all other models are adopted from their official implementation repository. For model details, refer to the original papers.

**Supervised (sup) static net** is the spatial stream of the hidden two-stream convolutional neural network model [Zhu et al., 2019]. The sup static net has a resnet18 architecture and encodes static

features of visual stimulus. Two versions of the model were included in the models' pool: one is trained on Image-Net [Deng et al., 2009] and predicts object identity, and the other is trained on HAA-500 action dataset [Chung et al., 2021] and predicts action label. Both versions take a single frame as input.

**Self-supervised (s-sup) dynamic net** is the first part of the temporal stream (i.e., motion net) in the hidden two-stream convolutional neural network model [Zhu et al., 2019]. The self-supervised dynamic net takes 11 consecutive frames as input and infers the optic flow between each pair of consecutive frames. The network is trained to minimize an self-supervised learning objective obtained by combining three loss functions:1) a pixel-wise reconstruction error, 2) a smoothness loss addressing the ambiguity problem of optic flow estimation (also known as the aperture problem), and 3) a structural dissimilarity between the original and the reconstructed image patches (see Zhu et al. [2019] for details of loss functions). The models' pool contains one version of the self-supervised dynamic net, trained on the HAA-500 action dataset Chung et al. [2021].

**Supervised (sup) dynamic net** is the second part of the temporal stream in the hidden two-stream convolutional neural network model [Zhu et al., 2019]. The model has resnet18 architecture and takes optic flows from the self-supervised dynamic net as input. We used the HAA-500 dataset Chung et al. [2021] and trained the supervised dynamic net to predict action labels using optic flows.

**Masked Autoencoder (MAE)** learn image representations, required to reconstruct original uncorrupted images from corrupted (masked) input through a series of transformer blocks He et al. [2022]. The models' pool contains two versions of the MAE model: 1) a pre-trained version, where the model is trained to reconstruct pixel values, and 2) a fine-tuned version, where the pre-trained model is further fine-tuned to predict object identities. Both versions were trained on Image-net Deng et al. [2009].

**Video Masked Autoencoder (VMAE)** learns a spatiotemporal representation of videos, required to reconstruct original uncorrupted videos, from corrupted (tube masked) input through a series of transformer blocks Tong et al. [2022]. We added two versions of the VMAE to our models' pool. The first is a pre-trained version, where the model is trained to reconstruct missing pixels of the input set of frames. The second version is the fine-tuned version obtained by fine-tuning the pre-trained version to predict action labels of input videos. Both models take a consecutive set of frames as input, and were trained on the Kinetics-400 action dataset Kay et al. [2017].

**Masked Video Distillation (MVD)** learns a higher-level spatial and spatiotemporal representation of the input video, required to reconstruct the representation of teacher MAE and VMAE while taking corrupted (tube-masked) videos as input [Wang et al., 2023a]. Unlike VMAE and MAE, the MVD model does not learn pixel-level features. Rather, it learns high-level features of the input video using pre-trained MAE and VMAE models' features as masked prediction targets. Using the Kinetics-400 action dataset Kay et al. [2017], a pre-trained version was obtained and added to the models' pool.

## 2.5 Models' Representational Dissimilarity Matrices (RDM)

In order to compare the models and the fMRI data, we computed representational dissimilarity matrices (RDMs) for the models' layers with a multi-step procedure. First, since the temporal resolution of the models' representations (25Hz) is much higher than the temporal resolution of fMRI data, we down-sampled each layer's activation timecourses over time by selecting one data point every five time points(down to 5 Hz). Then, we convolved the layer's activations with a standard Hemodynamic Response Function (HRF). Given that the fMRI data's repetition time (TR) is 2 seconds, we took a layer's activation every $25 \times 2 = 50$ time points.

Finally, for each layer we computed the dissimilarities between all pairs of timepoints, obtaining RDMs in which the entry at column $j$ and row $i$ contains correlation dissimilarity (1-Pearson's r) between the layer activations at time $i$ and time $j$. We repeated this procedure for BOLD responses to all eight movie segments, resulting in eight RDMs.

## 2.6 Brain Representational Dissimilarity Matrices (RDM)

RDMs were constructed separately for each brain stream in the subject's native space. The voxels for each brain stream were obtained as the union of the region voxels for individual regions within that stream. For each brain stream, we calculated the correlation dissimilarity ($1 - r$ where r is Pearson's correlation) of fMRI response patterns for all pairs of TRs. This yielded eight RDMs, corresponding to BOLD responses in eight video segments.

## 2.7  Measuring models similarity with brain data

To evaluate how well each model accounts for the activity in the brain streams, we used a cross-validated linear regression to predict the left-out movie segment brain stream RDM and computed the correlation between the predicted and the true RDM in each brain stream. The correlation captures how well a model's layers can predict a brain stream's responses to the visual stimuli. First, we used each model's layers' RDMs corresponding to seven (out of eight) video segments to train a linear regression model that predicts the corresponding seven RDMs in each brain stream. Then, we averaged the linear regression model's coefficients along the seven segments and used the averaged coefficients to predict the brain stream RDM of the left-out segment, using the model layers' RDMs of the corresponding segment. Finally, we calculated the Pearson's correlation between the predicted and the true RDMs. We repeated the leave-one-out cross-validation process for all the segments and averaged over the obtained correlations.

## 2.8  Measuring combined models similarity with brain data

We sought to study whether a combination of features from two models can improve similarity with brain data. We followed the procedure in 2.7 and used RDMs of all the layers in a pair of combined models to estimate the coefficients of a linear regression model that best predicts the RDM of a brain stream in seven (out of eight) of the video segments. Using leave-one-out cross-validation, we predicted the brain stream RDM of the left-out video segment using the average of the coefficients obtained from the seven video segments during training. Finally, we measured the correlation between the predicted RDM and the actual brain stream RDM to measure the correspondence between the combined models' features and the brain activity.

## 2.9  Measuring unique and shared similarity of a pair of models with brain data

To evaluate how well unique and shared features among a pair of computational models correspond to the brain data, we used Pearson's r to measure the accuracy of a "target" model's layers prediction of a brain stream RDM while controlling for the variation of a "control" model layers. Using leave-one-out cross-validation, first, we estimated the coefficients of a linear regression model that predicts a brain stream's RDM from the control model's layers in training video segments (seven out of eight). Second, we subtracted the predicted from the actual brain stream RDM in the training and the left-out video segments to obtain training and left-out residuals. Third, we estimated the coefficients of a linear regression model that predicts training residuals of each video segment using the target model layers. Finally, we measured Pearson's correlation between the target model's prediction of the left-out video segment residuals and the residuals obtained from the prediction of the control model.

# 3  Results

The human visual system does not consist of a single processing stream. Instead, it is organized into distinct neural pathways. To the extent that the structure of the human visual system is shaped by computational optimality, understanding the visual representations encoded in these pathways can offer insights into more general principles of vision. The contribution of this work is to quantify the similarity between the representations in the different visual pathways in the human brain and representations in models of vision that are widely used in Computer Science but understudied in Cognitive Neuroscience (e.g. masked autoencoders, two-stream networks, masked video distillation). In a first set of analyses 3, we leverage differences between models to reveal differences between the information encoded in different visual pathways in the brain. In a second set of analyses, we quantify the unique contribution of different deep network models to account for neural responses 3.2.

## 3.1  DNN models similarity with human brain streams

While numerous research studies have been conducted on model-to-brain correspondence using static images [Rose et al., 2021, Doshi and Konkle, 2023, Tsao et al., 2006], the impact of dynamic information on neural responses to naturalistic videos is understudied. To fill this gap, we tested the correspondence between neural representations in different visual pathways (ventral, dorsal,

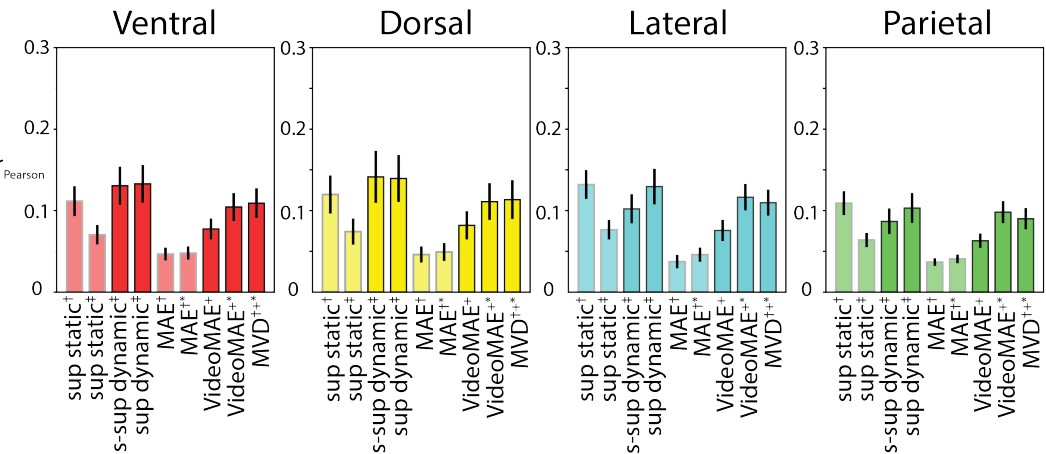

Figure 2: Pearson's correlation between actual and predicted brain stream RDMs, averaged over participants. Predicted RDMs were obtained by training and test a leave-one-out cross-validation linear regression model using each model's layers. Error bars show standard deviation over participants. Lighter bars correspond to models containing static, and darker ones to models containing dynamic visual information (sup: supervised, s-sup: self-supervised, †: Image-net-trained, ‡: HAA-500-trained, +: Kinetics-400-trained, *: fine-tuned; MVD was trained on pre-trained MAE (Image-net) and VideoMAE (Kinetics-400))

lateral, and parietal) and deep network models that can process dynamic information (two-stream networks, video masked autoencoders, and masked video distillation). Comparing the correspondence of neural responses with these models and their correspondence with models that only process static information (standard convolutional ResNets, masked autoencoders) made it possible to study the contribution of dynamic information independently of whether the learning objective is supervised (as in two-stream networks) or unsupervised (as in masked autoencoders). In addition, the study of the correspondence between neural representations and representations in masked autoencoders is of interest in its own right: masked autoencoders are effective and widely used, but little is known about their similarity to neural representations.

### 3.1.1 Static and dynamic information in brains and feed-forward convolutional neural networks (CNN)

Functional MRI responses recorded during the observation of naturalistic videos in the ventral, dorsal, lateral and parietal visual pathways were compared to the representations in feed-forward convolutional neural networks. The same dataset (HAA-500) was used to train the different branches of a hidden-two-stream network: the "supervised static" branch (a ResNet that takes as input individual frames of a video and computes as output the action category), the "unsupervised dynamic" branch (a convolutional network trained to compute optic flow by minimizing a self-supervised loss), and the "supervised dynamic" branch (a ResNet that takes as input optic flow and computes as output the action category). In addition, to facilitate parallels with prior work, we compared neural responses to a widely studied feed-forward model: a ResNet trained with Image-net.

Comparing deep network models trained with the same dataset (HAA-500) showed that models including dynamic information correlated with neural responses more than the Spatial model, that does not use dynamic information (Figure 2). This effect was observed for all visual pathways. In addition, representations in the lateral and parietal pathways correlated more with the supervised dynamic model than with the unsupervised dynamic model (fisher-transformed t-values with Bonferroni-corrected threshold). Lateral and parietal regions are located downstream compared to the dorsal regions, thus this result is complementary to earlier work that reported a correspondence between subsequent stages of processing in deep neural networks and in neural pathways in the case of static visual stimuli [Khaligh-Razavi and Kriegeskorte, 2014] and in the case of auditory stimuli [Kell et al., 2018].

Supervised CNNs trained with Image-Net performed well, achieving correspondence with neural responses that was close to that of HAA-trained models that included dynamic information. This could indicate that some of the variance in neural responses that correlates with dynamic models might also be accounted for by models trained exclusively with static information, as long as a suitable training dataset is used (in this case, Image-Net). However, an alternative possibility is that the supervised static model trained with Image-Net and the dynamic models trained with HAA might account for different portions of the variance in neural responses. We investigate these alternative possibilities in section 3.2.

### 3.1.2 Static and dynamic information in brains and masked autoencoders

Masked Autoencoders (MAE, [He et al., 2022]) and Video Masked Autoencoders (VideoMAE, [Tong et al., 2022, Feichtenhofer et al., 2022]) models are trained to reconstruct masked pixels of input (image or video) during pre-training and are further fine-tuned to predict object/action labels. MAE and VideoMAE models are very effective in learning visual representations and have been shown to outperform competing models in several visual tasks [He et al., 2022, Tong et al., 2022, Feichtenhofer et al., 2022, Wang et al., 2023b, Venkatesh et al.]. However, it is still unknown whether the representations learned by models based on masked autoencoding are similar to visual representations in the human brain. Here we investigated this question, quantifying the correlation between neural responses measured with fMRI while participants watched naturalistic videos, and representations learned by models trained with masked autoencoding.

We compared the correspondence between neural responses and MAEs trained with images (which learn spatial relationships between component of an image, [Wang et al., 2023a]) as well as Video-MAEs (which learn temporal relationships in videos, [Wang et al., 2023a]). Finally, we also compared neural responses to masked video distillation (MVD, [Wang et al., 2023a]), which combines image MAEs and videoMAEs to better capture both spatial and temporal relationships. Unlike MAE and VideoMAE, the MVD model does not aim to reconstruct missing patches at the level of pixel values. Instead, MVD adopts a knowledge-distillation approach, reconstructing missing information at the level of features extracted from pre-trained MAE and VideoMAE teachers.

As in the case of supervised models trained with the HAA dataset, models that included dynamic information (VideoMAEs) outperformed models using only static information (Image MAEs). This pattern was observed across all visual pathways. Image MAEs did not correlate well with neural responses, even compared to supervised models trained with static inputs. Overall, the representations learned by Image MAEs were very different from neural representations. By contrast, VideoMAEs showed greater correspondence with neural responses. In particular, fine-tuning with an action recognition task (Figure 2, VideoMAE fine-tuned) improved the correspondence between Video-MAE representations and neural representations across all streams (fisher-transformed t-values with Bonferroni-corrected threshold). Across all the pre-trained models, pre-trained MVD showed the highest similarity to neural representations in all brain streams. Further, MVD showed comparable similarity with brain streams to that of fine-tuned VideoMAE.

### 3.2 Vision models capture shared and unique neural activity variation in human brain streams

The results described in 3 show that representations from models trained with dynamic information are more correlated with neural representations compared to representations from models trained with static information. In particular, a mixed-effects model was conducted to examine the effect of model type (static vs. dynamic) on Fisher-transformed correlation values across subjects. Results indicated a significant effect of model type with models that process dynamic information having higher correlation (F(fixed effect df=1, residual df=502)=-13.487). This overall pattern is broken by the exception of ResNets trained with ImageNet, which performed on par with models trained with dynamic information. This raises the question of whether ResNets trained with ImageNet and dynamic models explain overlapping variance in neural responses or whether, instead, they are complementary, capturing non-overlapping portions of the variance. This question can be posed more generally for any pair of models studied in section 3. We investigated this first by combining layers from two models and measuring whether a combination of models can better predict the pattern of neural activity in visual pathways. Second, we measured the correspondence between a "target" model's representations and the representations in each brain stream while controlling for the

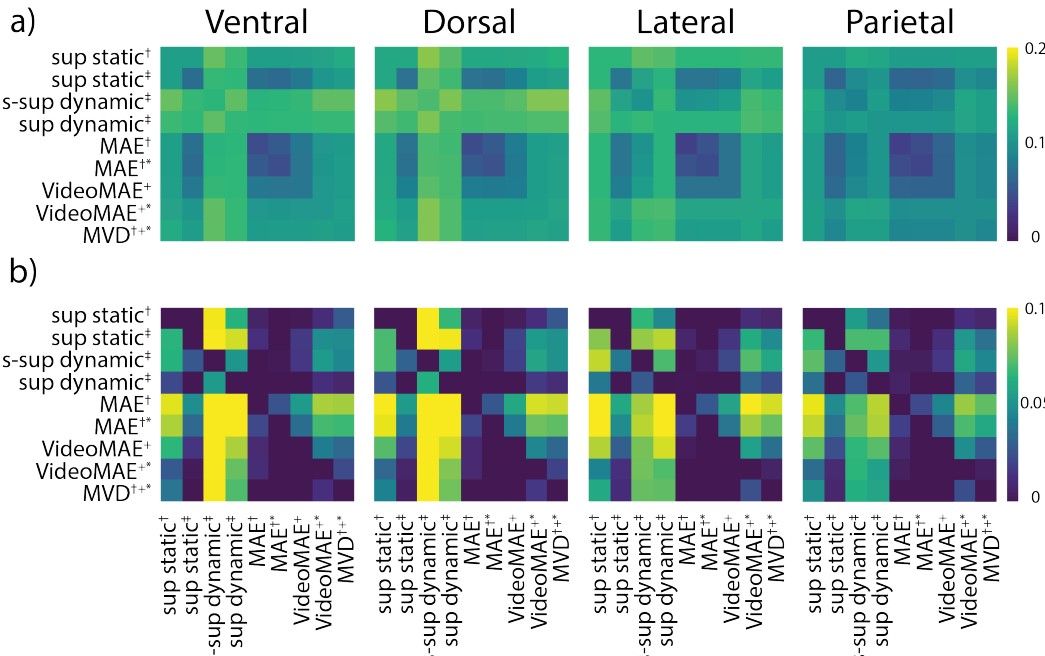

Figure 3: **a)** Model combination similarity with brain streams. The similarity was calculated using Pearson's correlation between a brain stream's actual and predicted RDMs. These predictions were obtained by combining layers from two models (corresponding to the row and column names), and averaged across participants. A linear regression model was trained and tested using leave-one-out cross-validation to generate the predictions.. **b)** Models unique similarity with brain streams. The similarity was calculated using Pearson's correlation between the actual RDM of a brain stream and the RDM predicted by a target model while controlling for the variation explained by a control model in the brain stream. These correlations were averaged across participants. Each row corresponds to a different control model, and each column corresponds to a different target model used for prediction. (sup: supervised, s-sup: self-supervised, †: Image-net-trained, ‡: HAA-500-trained, +: Kinetics-400-trained, *: fine-tuned; MVD was trained on pre-trained MAE (Image-net) and VideoMAE (Kinetics-400))

representations encoded in a "control" model. To this end, we predicted neural representations using the representations of the control model and obtained the residuals. Then, we predicted the residuals using the representations in the target model (see Methods for details).

Each matrix in Figure 3.a shows how well a combination of models' layers can predict the pattern of neural activity in a brain stream. Each column of each row demonstrates the correlation between the neural response pattern of a brain stream and the combined models' layers' prediction of that brain stream's neural activity pattern. Model-to-brain-stream similarity increased in all brain streams when combining features from static models with features from dynamic models. Notably, the correspondence between combined models' features with both dorsal and ventral streams improved in two cases: 1) combined features from Image-net-trained static supervised models with dynamic features from self-supervised model and 2) features from the combination of the self-supervised dynamic model with either VideoMAE or MVD. These cases shows that, first, ventral and dorsal brain streams both represent static and dynamic visual features, and second, different types of dynamic information are represented in both of these human streams.

Figure 3.b demonstrates the correspondence between a target model's features and each brain stream when we controlled for the features of a control model in the brain stream's neural responses. The results are visualized as a matrix in which each row corresponds to a control model and each column to a target model. The first row displays the correlations between models and neural responses after controlling for the Image-net-trained static model. The high values for the columns corresponding to the self-supervised dynamic and the supervised dynamic models indicate that these models and the Image-net-trained static model capture non-overlapping variance in neural responses. Representations

learned by these models also capture non-overlapping variance with those learned by the unsupervised dynamic models: the VideoMAEs. This finding shows that despite VideoMAEs exhibit relatively high correlations with neural responses (outperforming Image MAEs), they nonetheless fail to capture some variance in human visual representations that is accounted for by self-supervised and supervised dynamic models.

VideoMAEs and MVD accounted for additional variance in neural responses compared to MAEs (as expected given the results in Figure 2) but also compared to the HAA-trained static and self-supervised dynamic models. However, they accounted for a minimal amount (if any) of additional variance compared to the supervised dynamic model, suggesting some degree of convergence on common representations across models trained with different learning objectives.

## 4  Limitations

This study focused on a set of models selected to enable comparing the contribution of static and dynamic information and the impact of supervised and unsupervised learning objectives. The selection of models in this study includes only a subset of the existing models; future work will be needed to expand the set of models tested. In particular, future research should employ a larger space of models to disentangle the specific contributions of factors such as datasets, architectures, and loss functions to differences in brain predictability. In addition, the present work centered on the comparison between models and entire visual streams. A finer-grained analysis comparing models to individual regions within each stream will require further work.

## 5  Discussion

In the current study, we focused on predicting brain responses to a quasi-naturalistic movie using two popular classes of models—CNNs and MAEs—that also differ in their representation of spatial and temporal information. This yielded three main findings. First, models including dynamic information outperformed models using exclusively static information, not only in the dorsal, lateral, and parietal streams but also in the ventral stream. This is in line with recent evidence of responses to dynamic features in ventral brain regions [Robert et al., 2023]. Patients with deficits for motion perception typically present with lesions affecting dorsal regions (such as area V5, [McLeod, 1996, Vaina et al., 1990, Zihl et al., 1983]) or parietal regions [Battelli et al., 2003]. By contrast, patients with damage to ventral regions typically do not present with deficits for motion perception [Gilaie-Dotan et al., 2015]. This raises the question of what might be the use of dynamic information represented in the ventral stream. We hypothesize that this information might be used to support object segmentation, as proposed in recent computational models [Chen et al., 2022] inspired by classic work in Developmental Psychology [Spelke, 1990].

Second, Image MAEs showed little correspondence with neural representations, even compared to other models trained exclusively with static information. These results indicate that despite the effectiveness of Image MAEs for learning visual representations that can transfer to a variety of visual tasks [He et al., 2022], these models do not converge on representations that are similar to those observed in the human brain, suggesting that human vision and image MAEs rely on different computational mechanisms.

Third, models based on optic flow representations accounted for unique variance in all streams, even compared to video masked autoencoders that can make use of dynamic information. Fine-tuning video MAEs with an action classification task increased their correspondence with neural representations, but did not fully bridge the gap with neural responses compared to optic flow models, which still explained additional unique variance compared to the fine-tuned video MAEs. The additional contribution of optic flow models was particularly strong in ventral and dorsal regions, suggesting that representations based on optic flow exhibit greater correspondence with representations in early stages of visual processing in the human brain compared to both image and video MAEs.

In complex videos such as a movie, perceptual features and higher-level features can be partly correlated, thus, it is difficult to establish with certainty whether observed correlations between models and neural responses are driven by the neural representation of perceptual properties of the input or higher level information (see [Grall and Finn, 2022]). Nonetheless, it is possible to evaluate which models provide a better account of neural responses overall. Determining whether

the correspondence is driven by perceptual or higher-level features will require targeted experiments using stimuli in which these different types of features are uncorrelated.

The correlation values we obtained for our quasi-naturalistic stimuli (the movie Forrest Gump) are comparatively lower than those obtained in previous work for static images [Conwell et al., 2022], likely reflecting the greater complexity of the quasi-naturalistic stimuli. It also needs to be noted that the overall magnitude of the differences between static and dynamic models' correlation with brain responses is relatively small (approximately r=0.05).

The selection of models in this study includes only a subset of the existing models; future work will be needed to expand the set of models tested. In particular, future research should employ a larger space of models trained with a broader variety of datasets to disentangle the specific contributions of factors such as datasets, architectures, and loss functions to differences in brain predictability. Further comparing neural responses with models that compute motion energy will be an important direction for future research. In addition, research will be needed to test the ability of MAEs to account for responses in other regions, particularly the anterior portions of the temporal lobe (including the anterior superior temporal sulcus), which have been implicated in high-level and social perceptions.

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

# 6 Supplementary materials

## 6.1 Training and testing the Two-stream CNN for action recognition

We adopted the models in Zhu et al. [2019] and trained on the HAA500 dataset [Chung et al., 2021]. The dataset contains over 591k labeled frames with 500 action classes. 85% of the data points were used for training, 5% for validation, and 10% for testing 6.1. The training dataset was converted to the Webdataset format, i.e., shards of tar files. We used 4 V100 GPUs and 8 workers to load the dataset and train the models. All the analyses were performed on the same version of the movie that was used to acquire fMRI responses in the StudyForrest dataset [Hanke et al., 2016].
The *supervised static model* have a ResNet18 architecture [He et al., 2016], and were trained for 47 epochs with a batch size of 128. The training was done with the stochastic gradient descent algorithm with a 0.001 initial learning rate and a 0.0001 weight decay. During training, the gradients were accumulated and backpropagated for every two batches. Each frame in an input batch is a $224 \times 224$ frame and was randomly flipped horizontally.
The *unsupervised dynamic model* was trained for 12 epochs with a batch size of 32 and an initial learning rate of 0.01. No weight decay was used during training. Input to this model consists of a set of 11 frames each with dimensions of $224 \times 224$.
The *supervised dynamic model* was trained for 50 epochs with a batch size of 128 and an initial learning rate of 0.001. A weight decay of 0.0005 was used to train the models, and the gradients were accumulated and backpropagated every 5 batches.

Table 2: Test performance of models on the HAA500 dataset

| Model | epochs | Performance | |
| --- | --- | --- | --- |
| | | Top-1 | Top-3 |
| sup static | 47 | 30.80% | 49.38% |
| unsup dynamic + sup dynamic | 12, 50 | 22.72% | 37.90% |

