# OpenReview forum: "Comparing Representations in Static and Dynamic Vision Models to the Human Brain"
_NeurIPS.cc/2024/Workshop/UniReps — UniReps_

### Official Review · Reviewer_LYVu · 2024-10-04
**An interesting study demonstrating the importance of studying dynamic vision**

**Rating:** 8
**Confidence:** 4

**Review:**

This is an interesting paper examining the correspondence between different DNNs and the human brain in dynamic, natural contexts. Most work in this domain has been done with static images, and extending to the temporal domain is an important area.

The results show that video models / temporal dynamics are important to match visual cortex resopnses. This corresponds with other recent work with video clips (Garcia et al. 2024 PsyArxiv). Interestingly, even a simple optic flow model can account for unique variance in the neural responses.

I think there are a couple of limitations that would help to acknowledge / discuss. First, a continuous Hollywood movie (Forest Gump) has inherent correlations between the perceptual and higher level features, as well as autocorrelations across the movie that can make the results difficult to interpret (for an interesting discussion see Grall and Finn 2022). Second, like many neuroAI papers only a handful of models are compared to the brain. While it is tempting to attribute these model differences to broad distinctions between models (e.g., dynamic vs static or training objective), they differ along many other factors such as architecture. The relatively small set of models considered can make it difficult to draw strong conclusions about the cause of model differences. One other small point is that more anterior regions of the lateral stream in the STS were not modeled.

Overall this is an interesitng paper that points towards the importance of benchmarking neuroAI in dynamic, naturalistic contexts.

---

> ### Author Response · Authors · 2024-11-06
>
> Thank you for bringing Grall and Finn (2022) to our attention. It is important to acknowledge that perceptual and higher-level features can correlate in complex movies such as the one we used in the present study. We added this sentence to the Discussion section to highlight this point:
>
> “In complex videos such as a movie, perceptual features, and higher-level features can be partly correlated, thus, it is difficult to establish with certainty whether observed correlations between models and neural responses are driven by the neural representation of perceptual properties of the input or higher level information (see Grall and Finn, 2022).”
>
> Despite this, it is possible to evaluate which models provide a better account of neural responses in different brain regions. We added this passage to reflect this:
>
> “Nonetheless, it is possible to evaluate which models provide a better account of neural responses overall. Determining whether the correspondence is driven by perceptual features or higher level features will require targeted experiments using stimuli in which these different types of features are uncorrelated.”
>
> The ability to only test a subset of models is a limitation of this study, and caution needs to be applied in the interpretation of the results. We added the following to the Discussion section:
>
> “The selection of models in this study includes only a subset of the existing models; future work will be needed to expand the set of models tested. In particular, future research should employ a larger space of models trained with a broader variety of datasets to disentangle the specific contributions of factors such as datasets, architectures, and loss functions to differences in brain predictability.”
>
> One aspect worth noting is that we made an effort to include models with similar architectures that differ in terms of their use of static vs dynamic information. For example, the supervised dynamic CNN and the action-trained static CNN both have ResNet architectures with the same number of layers, and they were both trained with the HAA dataset, but the static net used individual frames as input, while the supervised dynamic net used optic flow. Similarly, for masked autoencoders, we included both image-based and video-based MAEs. The pattern of results we observed showed a better performance of the dynamic models (the supervised CNN and the video-based MAEs) compared to their static counterparts. Nonetheless, we agree that testing a large pool of models with a larger space of datasets will be important to characterize more accurately what are the key elements that drive differences in the models’ correspondence to neural responses.
>
> We also added this passage to highlight that it will be interesting to investigate the anterior STS in future work:
>
> “In addition, research will be needed to test the ability of MAEs to account for responses in other regions, particularly the anterior portions of the temporal lobe (including the anterior superior temporal sulcus), which have been implicated in high-level and social perceptions.”

---

### Official Review · Reviewer_48ZF · 2024-10-05
**Comparing Representations in Static and Dynamic Vision Models to the Human Brain**

**Rating:** 7
**Confidence:** 4

**Review:**

This work investigates how well embeddings from deep neural network models trained using static and dynamic inputs predict neural responses to a naturalistic, dynamic movie stimuli. This is an important area to research since it tests the generalization of insights gained in research settings to how the brain processes stimuli that are closer to daily life. The study also compares self-supervised and supervised models, including Masked Autoencoders.

Pros:

- Good methods to compare brains and models, comparing to all three streams is well motivated by the cognitive neuroscience literature
- Compared across models trained with the same dataset

I liked the analysis where they controlled a model with another model to see what unique information each model provided to the neural prediction. It was comprehensive and clearly shows the inability of the MAE’s in predicting neural activity and how the optic flow models predict neural activity the best.

Cons:

- Overall differences between static and dynamic stimuli are extremely small, with differences of about r = 0.05 (which corresponds to the dynamic model explaining just .25% (r^2=0.0025) more variance than the static model) between paired static and dynamic versions of models. This is fine (and interesting!) to report, but concluding that dynamic models are better predictors than static models seems too strong and premature.
- While I liked the analysis looking for unique variance, it is difficult to interpret some of the results since we don’t know what the differences are due to for some of the pairs of models (could be different datasets or the architectures or the loss functions for example).

It would have been nice for the authors to provide more motivation or a framing for why they thought each of these models would be a good neural predictor, especially for the Masked Autoencoders, which is their novel contribution. They gave good background on how these models do well on visual tasks and they state that they are understudied in Cognitive Neuroscience, but that is not quite a satisfactory reason to study them.

The results could be improved by comparing the optic flow models to pymoten (motion energy extracted using a using a pyramid of spatio-temporal Gabor filters) to see if these models learn specific dynamic visual representations that benefit predicting neural activity in each of the three streams.

---

> ### Author Response · Authors · 2024-11-06
>
> >*Overall differences between static and dynamic stimuli are extremely small, with differences of about r = 0.05 (which corresponds to the dynamic model explaining just .25% (r^2=0.0025) more variance than the static model) between paired static and dynamic versions of models. This is fine (and interesting!) to report, but concluding that dynamic models are better predictors than static models seems too strong and premature.*
>
> Thank you for raising this point. We agree that the differences between static and dynamic stimuli are relatively small. We added the following passage to the Discussion to address this:
> “It needs to be noted that the overall magnitude of the differences between static and dynamic models correlation with brain responses are relatively small (approximately r=0.05).”
>
> This observation needs to be considered in the context of the overall correlation obtained with static models. For example, the correlation between the ImageNet-trained supervised static model and different brain streams is also relatively low (r~0.12), as noted in the previous point - possibly due to the complexity of the quasi-naturalistic stimuli. Therefore, a difference of 0.05 after adding dynamic information represents more than a 30% increase in correlation, while seemingly small, is meaningful in this setting. To further establish the robustness of this increase, we have added statistical analyses across participants. A linear mixed-effects model reveals a significant main effect of model type (temporal models > spatial models). We have included this new statistical analysis in the revised manuscript.
>
> >*While I liked the analysis looking for unique variance, it is difficult to interpret some of the results since we don’t know what the differences are due to for some of the pairs of models (could be different datasets or the architectures or the loss functions for example).*
>
> Thank you for raising this question. We think that answering this question would require testing a larger space of models. In this study, we focused on predicting brain responses using two popular classes of models—CNNs and MAEs—that also differ in their representation of spatial and temporal information. Further analyses to disentangle the specific contributions of factors such as datasets, architectures, and loss functions to differences in brain predictability will be left for future research. We have clarified this in the manuscript by adding the following passage to the Limitation section:
>
> “This study focused on a set of models selected to enable comparing the contribution of static and dynamic information and the impact of supervised and unsupervised learning objectives. The selection of models in this study includes only a subset of the existing models; future work will be needed to expand the set of models tested. In particular, future research should employ a larger space of models trained with a broader variety of datasets to disentangle the specific contributions of factors such as datasets, architectures, and loss functions to differences in brain predictability. ”
>
> >*It would have been nice for the authors to provide more motivation or a framing for why they thought each of these models would be a good neural predictor, especially for the Masked Autoencoders, which is their novel contribution.*
>
> Thank you for raising this important point. We have added the following to the Introduction to clarify the rationale for testing Masked Autoencoders:
> “Previous work has indicated that models yielding more accurate categorization performance also typically offer a more accurate prediction of neural responses (Yamins et al. 2024). MAEs are remarkably effective at categorization, making them promising candidate models of neural responses. Additionally, thanks to their transformer architecture, MAEs can capture larger-scale spatial dependencies in images, and masked video autoencoders can capture temporal dependencies. Human observers are able to represent such dependencies as well, thus in principle, MAEs could account for unique variance in neural representations compared to convolutional neural networks (CNNs).”
>
>
> >*The results could be improved by comparing the optic flow models to pymoten (motion energy extracted using a using a pyramid of spatio-temporal Gabor filters) to see if these models learn specific dynamic visual representations that benefit predicting neural activity in each of the three streams.*
>
> Thank you for mentioning this. We agree that motion energy models are an excellent candidate to be included in the model pool. We intend to extend the work and study these models in our future research. We have added a note on this point to the Discussion section:
>
> “Further comparing neural responses with models that compute motion energy will be an important direction for future research”.

---

### Official Review · Reviewer_wYRb · 2024-10-07

**Rating:** 7
**Confidence:** 4

**Review:**

- Summary:

  - The study characterizes the ability of a set of models to predict neural responses to natural videos as measured via fMRI. In particular the authors are interested in whether: models trained on video inputs are more predictive (in general, or in supposed motion processing regions) of neural activity, and whether masked autoencoders are strong models of neural responses (which thus far has not been systematically tested in the literature.
  - The initial experiment measures the neural predictivity of several models in several different brain regions. My high level takeaways are (1) there is surprisingly little variation in neural predictivity across ventral/dorsal/lateral/parietal regions (2) Image MAEs are poor models of neural responses to natural videos and (3) models trained on dynamic tasks/inputs are generally slightly more predictive than models trained on single frames/static tasks *given that the models have been trained using the same dataset and finally (4) the advantage of dynamic training is approximately the same size as the advantage of training on a large/diverse set of images rather than HAA-500 (i.e. the height differnce between the first and second columns of the left panel of fig 2 is similar to the height difference between 2 and 3).
  - Next the author's conduct two experiments to determine the degree to which different models are explaining either unique or overlapping portions of the neural response. First they measure how well neural responses can be predicted from *pairs of model responses* (so, if two models are predicting non overlapping portions of variance their paired predictivity will be larger than either of their individual predictivity scores). And second, they train models to predict the remaining response variability after conditioning on a second model (essentially by having model A predict the residual response left after predicting the response with model B).  These analyses show that (1) the ImageNet static supervised model is explaining a mostly distinct portion of variance to the dynamically trained models and (2) VideoMAEs improve on MAEs, but mostly by explaining a similar part of the neural response signal to the other (supervised action recognition and self-supervised optic flow prediction) dynamic objective/models considered.

- Strengths:
  - Originality/Significance: The considered question is very interesting and, to my knowledge not systematically answered in the current literature.
  - Rigor: Experimental details are explained in significant detail (likely enough to reproduce the core results). It is clear the author's made a significant effort to provide reasonably reliable estimates of predictivity and their associated uncertainties.
  - Clever Design of Controls: I think the experiments to determine whether static and dynamic models are predicting overlapping portions of response variance were interesting and important. This is a good example of why the field should move away from simply reporting predictivity scores!

- Weaknesses/Suggestions/Questions:
  - All of the models predictivities are quite low (mostly pearson's r varies betwen .05 and .15 in Fig. 2). Is there any reason we should expect this particular modality of fMRI to be much more difficult to capture with ANNs (for example I think the cRSA scores in [1] are most similar to what you are measuring and these typically have values ~0.3).
  - Many of the error bars are significantly overlapping in Fig 2. What gives? Also this is Std over subjects, but what about the variance within subjects (over CV splits)? In all, it would be good to provide more context for evaluating the statistical significance of some of the differences.

[1] Conwell, Colin, et al. "What can 1.8 billion regressions tell us about the pressures shaping high-level visual representation in brains and machines?." BioRxiv (2022): 2022-03.

---

> ### Author Response · Authors · 2024-11-06
>
> >*All of the models predictivities are quite low (mostly pearson's r varies betwen .05 and .15 in Fig. 2). Is there any reason we should expect this particular modality of fMRI to be much more difficult to capture with ANNs (for example I think the cRSA scores in [1] are most similar to what you are measuring and these typically have values ~0.3).*
>
> Thank you for this feedback and for pointing us to Conwell et al. (2022) for comparison. We agree that our model-to-brain predictivity values are lower than those typically observed in studies like theirs. However, we believe this discrepancy likely arises because our study differs in the prediction task. While Conwell et al. (2022) measure models’ ability to predict fMRI responses to static images of scenes (Natural Scenes Dataset), our work evaluates models’ predictivity for fMRI responses to quasi-naturalistic, dynamic stimuli—in this case, the full-length movie Forrest Gump. Given these differences in stimulus type, a lower model-to-brain correlation may be expected in our study. Naturalistic, dynamic stimuli introduce complex temporal dependencies that likely add noise to both the fMRI signal and the prediction task, making it more challenging for artificial neural network models to capture these responses as accurately as they would in studies using static stimuli.
>
> We have added the following passage to the discussion section to highlight this point:
>
> “The correlation values we obtained for our quasi-naturalistic stimuli (the movie Forrest Gump) are comparatively lower than those obtained in previous work for static images (Conwell et al. 2022), likely reflecting the greater complexity of the quasi-naturalistic stimuli.”
>
>
> >*Many of the error bars are significantly overlapping in Fig 2. What gives? Also this is Std over subjects, but what about the variance within subjects (over CV splits)? In all, it would be good to provide more context for evaluating the statistical significance of some of the differences.*
>
> Thank you for pointing this out. We agree that our analysis did not account for within-subject variance, which is indeed an important consideration for evaluating model predictivity. While we intend to leave the matter to future research, we will add more statistical analyses at the group level to evaluate brain response predictivity.
> Specifically, we have added the following analyses:
>
> “A mixed-effects model was conducted to examine the effect of model type (static vs. dynamic) on Fisher-transformed correlation values across subjects. Results indicated a significant effect of model type with models that process dynamic information having higher correlation (F(fixed effect df=1, residual df=502)=-13.487)”

---

### Decision · Program_Chairs · 2024-10-10

**Decision:**

Accept

**Comment:**

In light of the positive reviewers' feedback and relevancy of the submission, we are pleased to accept this paper for presentation at UniReps 2024. We kindly ask the authors to incorporate the reviewers' suggestions and feedback in the final camera-ready version of the manuscript.